# Regionalized Protein Localization Domains in the Zebrafish Hair Cell Kinocilium

**DOI:** 10.3390/jdb11020028

**Published:** 2023-06-16

**Authors:** Timothy Erickson, William Paul Biggers, Kevin Williams, Shyanne E. Butland, Alexandra Venuto

**Affiliations:** 1Department of Biology, University of New Brunswick, Fredericton, NB E3B 5A3, Canada; 2Department of Biology, East Carolina University, Greenville, NC 27858, USA

**Keywords:** hair cells, kinocilium, cilia, deafness, hearing loss, lateral line, transcriptomics, TU-tagging, zebrafish, ANKEF1, ODF3L2, SAXO2

## Abstract

Sensory hair cells are the receptors for auditory, vestibular, and lateral line sensory organs in vertebrates. These cells are distinguished by “hair”-like projections from their apical surface collectively known as the hair bundle. Along with the staircase arrangement of the actin-filled stereocilia, the hair bundle features a single, non-motile, true cilium called the kinocilium. The kinocilium plays an important role in bundle development and the mechanics of sensory detection. To understand more about kinocilial development and structure, we performed a transcriptomic analysis of zebrafish hair cells to identify cilia-associated genes that have yet to be characterized in hair cells. In this study, we focused on three such genes—*ankef1a*, *odf3l2a*, and *saxo2*—because human or mouse orthologs are either associated with sensorineural hearing loss or are located near uncharacterized deafness loci. We made transgenic fish that express fluorescently tagged versions of their proteins, demonstrating their localization to the kinocilia of zebrafish hair cells. Furthermore, we found that Ankef1a, Odf3l2a, and Saxo2 exhibit distinct localization patterns along the length of the kinocilium and within the cell body. Lastly, we have reported a novel overexpression phenotype of Saxo2. Overall, these results suggest that the hair cell kinocilium in zebrafish is regionalized along its proximal-distal axis and set the groundwork to understand more about the roles of these kinocilial proteins in hair cells.

## 1. Introduction

Our senses of hearing and balance rely on the function of sensory receptors in our ears called hair cells. Sensory hair cells are distinguished by hair-like projections from their apical surfaces that detect and transduce physical stimuli such as sound waves or head movements. Most of these projections are actin-filled microvilli (called stereocilia) that form the characteristic stair-cased hair bundles. However, at some point during their development, all vertebrate hair cells also feature a single microtubule-based projection called the kinocilium.

Cilia play prominent roles in sensory cell development and function from an evolutionarily diverse set of organisms [1,2]. In mechanosensory hair cells, the kinocilium can fulfill both sensory and structural roles, depending on the type of hair cell. Although kinocilia degenerate around the onset of hearing in mammalian cochlear hair cells, they nonetheless play a developmental role in establishing hair bundle polarity in the auditory system [3] and mediating Sonic Hedgehog signaling to instruct cochlear development [4,5]. In contrast to the transient kinocilia of cochlear hair cells, the kinocilium is a permanent and prominent feature of most other vertebrate hair cells in the auditory, vestibular, and lateral line organs, including the vestibular hair cells of mammals. In non-cochlear cells, it appears that kinocilia play a less pronounced role in establishing bundle polarity. However, they contribute to the structure and functional mechanics of the sensory hair bundle [6,7,8], as well as act as attachment points for inert masses such as otoliths and otoconia [9]. Kinocilial dynamics may also be involved in the repair and regeneration of cochlear hair cells [10]. As such, characterizing the functional components of the kinocilium is fundamental to our understanding of sensory hair cells.

Hair cell kinocilia possess several features that distinguish them from non-motile primary cilia as well as from motile cilia and flagella. Although “kino-” means “motion” and kinocilia usually possess the 9+2 arrangement of axonemal microtubules typically found in motile cilia [11], the hair cell cilium is non-motile, lacking the inner dynein arms and nexin links found in motile cilia [12]. In this respect, kinocilia are similar to some mammalian olfactory cilia, which also have a 9+2 axoneme but lack the dynein machinery to generate motion [13]. However, given the impressive diversity of vertebrate hair cells, it is possible that active movement is a feature of some kinocilia [14,15]. In some contexts, non-9+2 arrangements of microtubules have also been reported [10,12]. Among vertebrate hair cells, there is great diversity in kinocilial morphology—from the transient kinocilia of cochlear hair cells to the bulbed kinocilia of amphibian hair cells to the towering kinocilia of superficial lateral line neuromasts (and everything in between). However, little is known about the molecular bases of this diversity.

Sensorineural hearing loss can result from mutations in cilia-associated genes as part of a syndromic ciliopathy, such as reported for Alström, Bardet-Biedl, and Usher syndromes [16,17,18]. Cilia- and microtubule-associated genes are also implicated in nonsyndromic hearing loss, such as *CCDC50* [19], *MAP1B* [20], and *DCDC2* [21]. *PCDH15* and *CDH23* are also examples of cilia-associated Usher proteins, in which certain alleles result in nonsyndromic hearing loss [22,23,24]. There may be other cilia-associated genes waiting to be identified that contribute to auditory and/or vestibular function.

Zebrafish hair cells are genetically and molecularly similar to mammalian hair cells [25], allowing us to utilize the genetic tractability of zebrafish to characterize kinocilia-associated genes. In this study, we used thiouracil (TU)-tagging to identify hair cell-enriched transcripts in zebrafish and focused on three putative kinocilial genes within the dataset—*ankef1a*, *odf3l2a*, and *saxo2*—because of their potential involvement in hearing loss. Mouse *Ankef1* lies within the coloboma (*Cm*) deletion locus, whose phenotypes include auditory and vestibular deficits [26,27]. The Shared Harvard Inner Ear Database (SHIELD) reports that human *ODF3L2* and *SAXO2* lie within deafness loci *DFNB72* and *DFNA30*, respectively [28]. We made transgenic fish that express tagged Ankef1a, Odf3l2a, and Saxo2 proteins and demonstrate their distinct localization patterns within the kinocilium of zebrafish hair cells. Our results set the groundwork for future studies to understand the role of these proteins in kinocilial structure and function.

## 2. Materials and Methods

### 2.1. Animal Husbandry

Adult zebrafish were maintained using standard procedures [29]. Larval fish were grown at 28.5 °C on a 14:10 h light cycle in 100 mm Petri dishes containing E3 media (5 mM NaCl, 0.17 mM KCl, 0.33 mM CaCl_2_, and 0.33 mM MgSO_4_ buffered with NaHCO_3_).

### 2.2. Transgenic Fish Lines

The *Tg(myo6b:EGFP)vo68*, *Tg(myo6b:actb1-EGFP)vo8*, and *Tg(myo6b:yfp-Hsa.TUBA)idc16* transgenic lines have been previously described [8,30,31] and were gifted by the laboratory of Teresa Nicolson. For the TU-tagging experiments, we used a *Tg(myo6b:HA-UPRT-P2A-NLS-mCherry)* line that was similar to, but distinct from, the *Tg(myo6b:HA-UPRT-P2A-mCherry)* line used previously [32]. The *Tg(myo6b:ankef1a-emGFP)unb4*, *Tg(myo6b:odf3l2a-emGFP)unb5*, *Tg(myo6b:saxo2-emGFP)unb6*, and *Tg(myo6b:saxo2-mKate2)unb7* lines were generated for this study. For simplicity, we referred to these lines as Ankef1a-GFP, Odf3l2a-GFP, Saxo2-GFP, and Saxo2-mKate2 throughout the manuscript. Coding sequences were amplified using total RNA from 7 days post-fertilization (dpf) zebrafish using the SuperScript IV One-step RT-PCR kit (Thermo Fisher, Waltham, MA, USA, 12594025) and the primers listed in Appendix A. Entry and destination constructs were made by standard Tol2 Gateway cloning protocols [33].

### 2.3. Thiouracil (TU)-Tagging and Bioinformatics

TU-tagging experiments were performed essentially as previously described [32] with the following modifications. Transgenic and wild-type sibling larvae from outcrosses of a *Tg(myo6b:HA-UPRT-P2A-NLS-mCherry)* line were used (abbreviated as *Tg(UPRT)*). Stock solutions of 4-thiouracil (4TU; Sigma-Aldrich, St. Louis, MO, USA, 440736) and 4-thiouridine (4sU; Sigma-Aldrich, T4509) were prepared at 0.5 M and 0.25 M, respectively, in dimethyl sulfoxide (DMSO; Sigma-Aldrich, D8418). Working solutions were diluted in E3 medium with a final DMSO concentration of 0.5%. At 4 dpf, *Tg(UPRT)* larvae were treated with 2 mM 4TU, whereas non-transgenic siblings were treated with 1 mM 4sU. Both groups were bathed in their respective compounds for 3.5 h at 29 °C. Conversion of 4TU to 4-thiouridine monophosphate is enhanced in the presence of the *Toxoplasma gondii* UPRT enzyme, whereas endogenous biosynthetic pathways can produce 4-thiouridine monophosphate directly from 4sU. Using this strategy, TU-tagged RNA was preferentially synthesized in the hair cells of *Tg(UPRT)* larvae treated with 4TU, whereas RNA was tagged without bias for cell type in 4sU-treated, non-transgenic controls. The experiment was performed in a biological replicate, in which a biological sample was defined as 100 larvae per treatment group produced by a single breeding event.

Samples were processed in TRIzol reagent (Thermo Fisher, 15596026) according to the manufacturer’s protocol and enriched for polyadenylated mRNA using the Dynabeads Oligo(dT)25 kit (Thermo Fisher, 61005). Biotinylation of the mRNA was performed using the MTSEA-biotin-XX reagent (Biotium, Fremont, CA, USA, 90066). A 5 mg/mL solution was prepared in *N*,*N*-Dimethylformamide (DMF; Sigma-Aldrich, 227056). A 10X biotinylation buffer was prepared by diluting 1 M Tris-HCl pH 7.5 (Thermo Fisher, 15567027) to 100 mM and 0.5 M EDTA pH 8 (Thermo Fisher, 15567027) to 10 mM in nuclease-free water (Thermo Fisher, AM9930). The biotinylation reaction was performed by heating the mRNA in a 1X biotinylation buffer solution at 65 °C for 2 min before adding 10 µg of MTSEA-biotin-XX solution with a final DMF concentration of 25%. Samples were incubated in the dark at room temperature for 30 min on a rotating rack. mRNA was recovered by adding an equal volume of chloroform:isoamyl alcohol 24:1 (Sigma-Aldrich, C0549), vortexing, retrieving the aqueous phase with 5Prime Phase Lock Gel Heavy microfuge tubes (Quantabio, Beverly, MA, USA, 2302830), and performing a NaCl-isopropanol precipitation with glycogen (Thermo Fisher, R0551) acting as an inert carrier. Biotinylated RNA was fragmented to approximately 200–500 bases using the NEBNext Magnesium RNA Fragmentation Module (New England Biolabs, Ipswich, MA, USA, E6150S) and precipitated by adding 1/10 volume of 3M sodium acetate pH 5.5 (Thermo Fisher, AM9740) plus 3 volumes of 100% ethanol, again with glycogen as an inert carrier. Finally, biotinylated TU-tagged RNA was separated from untagged RNA using the µMacs Streptavidin kit (Miltenyi Biotec, Bergisch Gladbach, Germany, 130-074-101). High-salt MPG buffer (100 mM Tris, 10 mM EDTA, 1 M NaCl, 0.1% Tween-20) and 100 mM dithiothreitol (DTT; Sigma-Aldrich, 3860-OP) were prepared fresh. RNA samples (50 uL) were heated to 65 °C for 5 min and then iced. Resuspended µMass beads (100 µL) were added and the samples were incubated for 15 min in the dark at room temperature with rotation. µMacs columns were equilibrated with 2 × 100 µL of nucleic acid buffer (supplied with the kit) before applying the RNA-bead complexes to the columns. Columns were washed twice with 65 °C MPG buffer and twice with room temperature buffer. TU-tagged RNA was eluted from each column using 2 × 100 µL of 100 mM DTT pre-heated to 65 °C. The eluted RNA was recovered by NaOAc/ethanol precipitation, as described above, resulting in two biological samples of hair cell-enriched, TU-tagged RNA from *Tg(UPRT)* larvae treated with 4TU and two biological samples of tagged RNA from non-transgenic larvae treated with 4sU.

RNA-seq library construction and sequencing were performed by the OHSU Massively Parallel Sequencing Shared Resource (Oregon Health & Science University, Portland, OR, USA). Raw RNA-seq data were imported into an OHSU instance of the Galaxy platform [34] and mapped to the *Danio rerio* GRCz10 genome. The BAM files were imported to SeqMonk v1.48.1 (www.bioinformatics.babraham.ac.uk/projects/seqmonk/) for data visualization and statistical analysis using default parameters for DESeq2. Transcripts that were significantly enriched in the *Tg(UPRT)* samples were analyzed for GO biological process term enrichment using the Panther database (accessed on 19 December 2022) [35,36]. Looking for other possible microtubule-associated genes, we also manually annotated the enriched genes that did not have GO annotation by searching PubMed for “(‘*root gene symbol*’) AND (microtubule OR cilium OR cilia OR flagella OR axoneme)”. Analyses of *ODF3L2* and *SAXO2* genomic positions relative to known deafness loci were performed using information from the Hereditary Hearing Loss website [37] and by searching the GRCh38.p13 human genome assembly through Ensembl [38] for the loci-defining genetic markers described in the relevant publications.

### 2.4. Whole Mount mRNA In Situ Hybridization

mRNA in situ hybridizations (ISH) for *ankef1a*, *odf3l2a,* and *saxo2* were performed as described in [32]. Probe templates were amplified using the primers listed in Appendix A. The *odf3l2a* template was amplified from total RNA with T3 and T7 RNA polymerase sites on the forward and reverse primers, respectively, and the DIG-labeled antisense probe was synthesized directly from the purified PCR product using T7 RNA polymerase (Biolabs, M0251S). The *ankef1a* and *saxo2* templates were amplified from total RNA using gene-specific primers and cloned into the pCR4-TOPO vector (Thermo Fisher, 450071). DIG-labeled antisense probes for *ankef1a* and *saxo2* were synthesized from linearized plasmids using the appropriate RNA polymerase (*ankef1a*: T3 (New England Biolabs, M0378S); *saxo2*: T7). Note that our *ankef1a* probe encompasses the entire open reading frame (2340 bp), with 613 bp at its 3′ end overlapping with a previously published *ankef1a* probe [39].

### 2.5. Confocal Microscopy, Image Analysis, and Statistics

Live zebrafish larvae (5–7 dpf) were anesthetized with 0.016% MS-222 and mounted in 1.2% low-melting point agarose overlaid with E3 media containing MS-222. Confocal imaging was performed using a Zeiss LSM 800 confocal microscope (Jena, Germany) equipped with a 40× water-immersion objective. To image inner ear sensory patches, optical slices were acquired every 0.6–0.7 µm for a depth of 22–32 µm. For clarity, projections were made from a contiguous subset of the complete stack, typically representing depths of 7–20 µm. Neuromasts D1, MI1, O1, or SO_2_ were imaged. Because the fluorescence intensity levels differed between transgenic lines, the gain settings were tailored for each transgene. In particular, the *saxo2-mKate2* “severe” line was imaged at lower gain settings to avoid overexposing the intensely bright mKate2 puncta. Pixel intensity and kinocilium length analyses were done in Fiji [40] and the “cool” lookup table was used to encode pixel intensity in maximum Z-projections. Neuromast images in Figure 2E–H contain “choppiness” artifacts produced by the rotation of 3D Z-projections. Images were compiled and adjusted for brightness and contrast in Adobe Photoshop CC.

Length measurements for lateral cristae kinocilia were done by tracing a kinocilium from the base to the tip through a confocal Z-stack using the SNT plugin for ImageJ [41,42] and calculating the Euclidean distance for kinocilial length using the Pythagorean theorem. The transmitted light channel was used to define the start and end of each kinocilium. A single biological sample (n) was defined as the per-larva average of the five tallest kinocilia that could be unambiguously traced. Per-larva averages were used for statistical comparisons between the transgenic larvae and non-transgenic siblings.

Fluorescent protein distributions along the length of lateral crista kinocilia were quantified in ImageJ using the “plot profile” function with a segmented line width setting of “2”. The 16-bit pixel intensity values were recorded every 0.064 µm. Measurable kino-cilia were defined as having a minimum length of 32 µm, over which the intensity profile could be unambiguously measured from the base to the distal tip. To account for differences in kinocilial length, pixel intensity values were registered to the most distal measurement of each kinocilium, meaning that proximal intensity values for kinocilia longer than 32 µm were not included in the intensity profile measurement analyses. Pixel intensity values at each position were averaged from all measured kinocilia to create an average gray value over a distance of 32 µm. To account for differences in overall fluorescence levels between transgenes, the average gray values were rescaled using min-max normalization to a 0–100 scale and plotted as log2 values fitted with a Loess curve.

RNA sequencing data were visualized and plotted using Seqmonk v1.48.1 (https://www.bioinformatics.babraham.ac.uk/projects/seqmonk/) and analyzed using the default Seqmonk settings for DESeq2. Statistical tests for kinocilial length were performed in R Studio [43] using the base R stats package [44] for two-tailed Welch’s *t*-tests and ANOVA. Plots were generated with ggplot2 [45].

## 3. Results

### 3.1. Identification of Putative Kinocilia-Associated, Hair Cell-Enriched Transcripts

Thiouracil (TU) tagging is a method of purifying mRNA transcripts from specific cell types [46]. To identify genes expressed in sensory hair cells (Figure 1A), we performed TU-tagged RNA-seq experiments on larval zebrafish, introducing several protocol changes relative to our previous study (See Methods) [32]. We identified 1838 features that were significantly enriched in the 4TU-treated *Tg(UPRT)* sample vs. the 4sU-treated wild-type controls (Figure 1B). Appendix A has the full list of significantly TU-enriched genes in this study. Gene Ontology (GO) enrichment analysis of 1675 uniquely mapped features reveals that genes associated with hearing, hair cells, and inner ear development are statistically overrepresented in the TU-enriched dataset (Appendix A), confirming a successful enrichment of hair cell transcripts. There was also a significant enrichment of genes associated with microtubule- and cilium-based processes (Appendix A), highlighting genes with potential roles in the structure and/or function of kinocilia.

Next, we manually searched the entire TU-enriched dataset for putative kinocilial genes that may be associated with deafness phenotypes in mice and humans, but whose protein products have not been previously localized to the kinocilium. In this study, we focused on three such genes—*ankef1a*, *odf3l2a*, and *saxo2*. We used mRNA in situ hybridization to confirm that the expression of all three genes was enriched in hair cells. *ankef1a* was primarily expressed in the inner ear and lateral line hair cells (Figure 1C). This restricted pattern differs from a previous study, in which *ankef1a* was shown to be more broadly expressed [39]. Using a probe against the 3′ end of the *ankef1a* transcript, similar to the one used in the earlier study, resulted in a comparably broad expression pattern (Appendix A). *odf3l2a* expression is restricted to the inner ear and lateral line hair cells (Figure 1D), consistent with the pattern reported for the *et10.4* enhancer trap line where the transgenic insertion is 1.5 kilobases (kb) upstream of the *odf3l2a* gene [47]. *saxo2* is also expressed in otic and lateral line hair cells, with additional robust expression in the developing olfactory epithelia (Figure 1E).

### 3.2. Evaluation of ANKEF1, ODF3L2, and SAXO2 as Potential Human Deafness Genes

Ankyrin repeat and EF-hand domain containing 1 (ANKEF1) proteins are predicted to contain protein–protein interaction (ankyrin) and calcium-sensing/binding (EF-hand) domains. Functionally, *Ankef1* has been implicated in *Xenopus* gastrulation and fertility in male mice [48,49]. ANKEF1 protein localization to the cilium has not been demonstrated, but transcriptomic and proteomic analyses suggest that it is enriched in ciliated cells (including mouse and zebrafish hair cells) [39,50,51,52,53,54,55,56,57].

Human *ANKEF1* is on chromosome 20 (GRCh38: 20:10034987-10058303) and is not located near any known human deafness loci [37] (Table 1). However, the mouse *Ankef1* ortholog is located within the coloboma (*Cm*) deletion locus [58,59]. *Cm* heterozygotes exhibit morphological abnormalities in the organ of Corti and a head bobbing phenotype indicative of vestibular defects [27,60]. Most of these *Cm* phenotypes can be attributed to the heterozygous loss of the Notch ligand Jagged 1 (*Jag1*). However, the authors note that the *Cm* and *Jag1* mutant phenotypes are not equivalent and speculate that other genes within the deletion may contribute to inner ear defects in *Cm* mutants [27].

Outer dense fiber protein 3-like 2 (ODF3L2) is a member of the ODF3 family of proteins characterized by sperm-tail PG (proline-glycine)-rich (STPGR) repeat domains. Although ODF families 1–4 share a common nomenclature and associate with microtubule-based structures, these four ODF families are not evolutionarily related to one another. ODF3 (SHIPPO1) was first identified as a component of the outer dense fibers (ODFs) of mouse sperm tails [61]. However, *Odf3* family genes are also expressed in ciliated cells that lack dense outer fibers [62,63,64,65], suggesting a broader role in microtubule-based structures [66]. Consistent with this idea, an Odf3l2 homolog in *Giardia* forms a part of the basal body proteome [67], human ODF3L2 localizes to cytoplasmic microtubules in a heterologous expression system [68], and *Tetrahymena* ODF3L homologs bind to axonemal microtubules [69]. However, ODF3L2 localization in vertebrate cilia has not been demonstrated. Functionally, little is known about the ODF3 and ODF3L proteins in any cell type. Zebrafish *odf3l2a* is a hair cell-enriched transcript [57] and, although evidence for *Odf3l2* expression in mammalian hair cells is weak, *Odf3l2* knockout mice display high-frequency hearing loss and vestibular dysfunction [70]. The underlying cellular causes of these phenotypes are unknown.

Human *ODF3L2* is located on chromosome 19p13.3 (GRCh38:19:463346-474880) (Table 1). The 19p13 region is home to several hearing loss loci, including *DFNB15*/*72*/*95* (*GIPC3*) [71,72], *DFNB68* (*S1PR2*) [73], *DFNB81*/Perrault syndrome (*CLPP*) [71,74], as well as the autosomal dominant *DNFA57* locus (gene unknown) [75]. The *DFNB72* critical region was mapped between markers D19S216 and D19S1034 (GRCh38:19:4949401-6113471) [76]. Although the Shared Harvard Inner-Ear Database (SHIELD) [28] reports that *ODF3L2* is located within the *DFNB72* critical region, the GRCh38 human genome assembly places *ODF3L2* at least 4.4 MB away from the closest *DFNB72* marker D19S216. The *DFNA57* critical region was mapped between markers D19S912 and D19S212 (GRCh38:19:7806762:18232460). *ODF3L2* is almost 7 MB away from the closest marker D19S592 and over 9 MB away from the maximum LOD score marker D19S586. However, the telomeric boundary was defined from a single patient, and the authors acknowledge the unlikely possibility that the *DFNA57* interval extends to the telomere. If this is the case, *ODF3L2* remains a candidate gene for *DFNA57*.

**Table 1 jdb-11-00028-t001:** Human *ANKEF1*, *ODF3L2*, and *SAXO2* chromosomal locations compared to potentially associated deafness loci.

Human Gene	Cytogenetic Location	GRCh38.p13 Coordinates	Proposed Deafness Locus	Critical Region Markers	Locus Coordinates	Causative Gene	Reference
*ANKEF1*	20p12.2	20:10034987-10058303	n/a	n/a	n/a	n/a	n/a
*ODF3L2*	19p13.3	19:463346-474880	*DFNB72*	D19S216–D19S1034	19:4949401-6113471	*GIPC3*	Ain et al. [76]
			*DFNA57*	D19S912–D19S212	19:7806762-18232460	unknown	Bönsch et al. [75]
*SAXO2*	15q25.2	15:82262818-82284927	*DFNA30*	D15S151–D15S130	15:87053920-94168232	unknown	Mangino et al. [77]

Stabilizer of axonemal microtubules 2 (SAXO2/FAM154B) is the second of the two SAXO family members that are part of the microtubule-associated protein 6 (MAP6) family [78,79]. SAXO proteins can stabilize microtubules and use their “Mn” repeat domains to associate with basal bodies, centrioles, and cilia [79,80,81,82]. Interestingly, SAXO and MAP6 are MIPs (microtubule inner proteins), meaning that they reside in the luminal space of microtubules [83,84,85,86]. Furthermore, luminal MAP6 can stabilize microtubule curvature [87]; however, it is unknown whether SAXO proteins also share this functionality. *Saxo2* is enriched in hair cell transcriptomic datasets from mice and zebrafish [55,57,88], but it has not been functionally characterized.

Human *SAXO2* is located on 15q25.2 (Table 1) near several sensorineural deafness loci, including *DFNA30* (gene unknown) [77], *DFNA68* (*HOMER*) [89,90], *DFNB48* (*CIB2*) [91,92], and *OTSC1* [93]. SHIELD reports that *SAXO2* maps to the *DFNA30* locus. The critical region for *DFNA30* lies between markers D15S151 and D15S130 (GRCh38:15:87053920-94168232), with a maximum LOD score for marker D15S1004 [77]. However, current genomic coordinates for *SAXO2* (GRCh38:15:82262818-82284927) place it outside the *DFNA30* critical interval, 4.8 MB away from the closest marker D15S151 and 11.5 MB away from D15S1004.

### 3.3. Distinct Localization Patterns of Ankef1a, Odf3l2a, and Saxo2 in the Kinocilium

To analyze the subcellular localization of Ankef1a, Odf3l2a, and Saxo2 in zebrafish hair cells, we created stable transgenic fish expressing fluorescently tagged versions of these proteins under the control of the hair cell-specific *myo6b* promoter (Figure 1F). Like YFP-Tubulin, tagged Ankef1a, Odf3l2a, and Saxo2 proteins all exhibit robust localization to kinocilia (lateral crista at 5–6 dpf; Figure 1G). In comparison, EGFP alone is largely restricted to the cytoplasm and nucleoplasm, with a faint signal in kinocilia (Appendix A).

We imaged the lateral cristae and neuromasts of live larvae at 5–6 dpf and encoded differences in pixel intensity by applying the ImageJ “cool” lookup table to maximum z-projections (Figure 2(A1–D1),E–H). We also quantified the fluorescence intensity in kinocilia of the lateral cristae and presented the average fluorescence intensity distribution as a log2 mean grey value over the most distal 32 µm (Figure 2(A2–D2)). For comparison, we also used the *Tg(myo6b:yfp-Hsa.TUBA)idc16* line that expresses YFP-tagged alpha-tubulin in hair cells, marking both cytoplasmic and axonemal microtubules as well as a pool of unpolymerized tubulin. As expected, YFP-TUBA is uniformly distributed along the length of the hair cell kinocilium in the lateral cristae and neuromasts (Figure 2(A1,A2),E). Ankef1a-GFP displays a similarly uniform distribution from the base to the tip (Figure 2(B1,B2),F). The distribution of Odf3l2a-GFP is distinct from that of tubulin and Ankef1a, with Odf3l2a being more restricted to the middle-proximal region of the kinocilium (Figure 2(C1,C2),G). The distribution of Saxo2-mKate2 is complementary to that of Odf3l2a, with enriched localization to the distal region of the kinocilium and depletion in the middle-proximal zone (Figure 2(D1,D2),H). Interestingly, we observed bent kinocilia in the lateral cristae of Saxo2-mKate2 transgenics (Figure 2(D1)), a phenotype we did not observe in the other transgenic lines or in non-transgenic siblings. Saxo2-GFP is also enriched in the distal region (Appendix A), but we did not observe a kinocilial curvature in this line. All five tagged proteins also show similarly distinct localization patterns in the hair cells of the anterior macula (Appendix A). Together, these transgenes demonstrate that Ankef1a, Odf3l2a, and Saxo2 are cilia-associated proteins and suggest that each exhibits a distinct distribution pattern in the kinocilia of zebrafish hair cells.

### 3.4. Ankef1a, Odf3l2a, and Saxo2 Exhibit Distinct Localization Patterns in the Hair Cell Soma That Differ from Cytoplasmic Tubulin

To determine whether Ankef1a, Odf3l2a, and Saxo2 are also associated with cytoplasmic tubulin, we compared the distributions of these tagged proteins to that of YFP-TUBA in the soma of hair cells in the lateral cristae and neuromasts (Figure 3). YFP-TUBA signal is strongest in the apical subcuticular zone that lies just below the actin-rich cuticular plate (Figure 3A). Cytoplasmic microtubules are also evident. Ankef1a-GFP is largely excluded from the subcuticular zone and there is no obvious association with the cytoplasmic microtubules (Figure 3B). Instead, the Ankef1a-GFP signal is diffused throughout the perinuclear cytoplasm. In contrast to Ankef1a, the Odf3l2a-GFP signal is mostly absent from the cytoplasm and accumulates in the nucleus of some hair cells (Figure 3C). Ankef1a and Odf3l2a display the same cytoplasmic localization patterns in neuromasts as they do in cristae (Appendix A). Lastly, Saxo2-mKate2 does not associate with cytoplasmic microtubules but instead accumulates in the subcuticular zone beneath the cuticular plate, marked here by GFP-tagged beta-actin (Figure 3D–F). YFP-TUBA also accumulates in this region (Appendix A). Likewise, Saxo2-GFP displays similar apical accumulation in the cell body (Figure 1G and Appendix A). In the top-down views of neuromast hair cells, both Saxo2-mKate2 (Figure 3F) and Saxo2-GFP (Appendix A) accumulate in the ring-like structures surrounding the cuticular plate. In summary, although Ankef1a, Odf3l2a, and Saxo2 all localize with tubulin in the kinocilium, none of these proteins share the same distribution pattern as YFP-TUBA in the hair cell soma, nor do their patterns overlap with one another.

### 3.5. Saxo2-mKate2 Overexpression Results in Curved, Truncated, and ‘Bulbed’ Kinocilia

Next, we measured kinocilial height in the lateral crista to determine whether overexpression of Ankef1a, Odf3l2a, or Saxo2 influences kinocilial morphology. The average maximum heights of kinocilia in the *YFP-TUBA*, *ankef1a-GFP*, and *odf3l2a-GFP* transgenics were statistically indistinguishable from their non-transgenic siblings (Figure 4A). In contrast to these single transgenic lines, we identified at least three independent lines for *saxo2-mKate2* with different levels of expression (Appendix A), which we categorized as “mild”, “intermediate”, and “severe” in terms of their kinocilial phenotypes (Figure 4B–E). A one-way ANOVA comparing the three *saxo2-mKate2* phenotypes and non-transgenic siblings reveals significant differences in kinocilial length (F3,24 = 193.3, *p* < 2 × 10^−16^). Compared to non-transgenic siblings, maximum height is not affected in “mild” *saxo2-mKate2* transgenics, despite examples of curved kinocilia (Figure 4A–C; non-Tg, n = 8, mean = 35.4 µm; “mild”, n = 8, mean = 35.3 µm; padj = 0.9858). However, both the “intermediate” and “severe” phenotypes have significantly shorter kinocilia compared to non-transgenic siblings (Figure 4A,D,E; “intermediate”, n = 7 mean = 32.1 µm, padj < 0.0001; “severe”, n = 5, mean = 26.1 µm, padj < 0.0001). Transgenics with the “intermediate” phenotype exhibit kinocilial curvature and bulbed tips marked by an accumulation of Saxo2-mKate2 (Figure 4D). Lastly, the “severe” Saxo2-mKate2 overexpression phenotype includes a further reduction in average maximum kinocilia height and increased prominence of bulbed tips (Figure 4E). The severe phenotype is also marked by a punctate accumulation of Saxo2-mKate2 in the cell bodies and along the kinocilia. Some of these puncta are bidirectionally mobile in the kinocilia (Appendix A). Neuromasts from severely affected larvae display a similar phenotype of truncated kinocilia (Appendix A). Surprisingly, these “severe” *saxo2-mKate2* transgenics are viable and qualitatively do not exhibit obvious auditory or vestibular phenotypes as larvae or adults. The Saxo2-GFP line does not exhibit curved kinocilia or a significant change in kinocilial height (t = −1.8712, df = 5.5674, *p*-value = 0.1143) (Appendix A). In summary, the transgenic expression of GFP-tagged Ankef1a, Odf3l2a, and Saxo2 does not affect kinocilial height. This is also true for the Saxo2-mKate2 line with lower levels of expression, although curved kinocilia are common. Transgenic lines with elevated Saxo2-mKate2 have bulbed and truncated kinocilia.

### 3.6. Ankef1a, Odf3l2a, and Tubulin Distribution Patterns Are Not Influenced by Saxo2

Given that Saxo2-mKate2 can alter kinocilial morphology, we next tested whether the presence of Saxo2-mKate2 influences the distributions of YFP-TUBA, Ankef1a-GFP, Odf3l2a-GFP, and Saxo2-GFP. We produced four combinations of double transgenic fish, each with Saxo2-mKate2 as a common factor. The co-expression of Saxo2-mKate2 with YFP-TUBA (Figure 5(A1–5) and Appendix A), Ankef1a-GFP (Figure 5(B1–5)), or Odf3l2a-GFP (Figure 5(C1–5)) does not alter the distribution patterns of either pair of proteins relative to single transgenic siblings. An example of the Saxo2-mKate2 “intermediate” phenotype is shown in panels B1, B3–5 while the “mild” phenotype is shown in the other examples. On the other hand, rather than simply overlapping with one another, the co-expression of Saxo2-mKate and Saxo2-GFP reveals a form of competition for space between these two transgenic proteins (Figure 5(D1–5)). Most obviously at the tips, either the mKate2 or GFP version of Saxo2 is more abundant, with Saxo2-mKate2 appearing to outcompete Saxo2-GFP at the tips of most kinocilia. Taken together, these data suggest that Saxo2 does not positively or negatively influence the distributions of tubulin, Ankef1a, or Odf3l2a in the kinocilium.

## 4. Discussion

### 4.1. Utility of TU-Tagging to Identify Hair Cell Transcripts

As in our previously published work [32], in this study we showed that hair cell-enriched transcripts can be identified by thiouracil tagging of newly synthesized RNA. GO analysis confirms that hair cell transcripts are overrepresented in the TU-tagged samples. We also identified all of the TU-enriched transcripts from our previous report, although some gene identifiers from GRCz10 are no longer part of the current GRCz11 genome assembly (*stereocilin-like* genes ENSDARG00000078845 and ENSDARG00000088059, *fam188b2* ENSDARG00000069867, uncharacterized gene ENSDARG00000088304). Changes to the experimental protocol greatly increased the number of significantly enriched genes in the TU-tagged samples. However, in addition to the enrichment of genes related to hair cells and cilia, this dataset is also enriched for genes involved in electron transport and ATP synthesis, iron metabolism, and visual system development and function (Appendix A). These results indicate a signal-to-noise ratio problem that seems inherent to TU-tagging in zebrafish. Unless one’s experiment requires time-resolved differentiation between nascent and preexisting pools of RNA, recent advances in single-cell RNA sequencing have rendered more error-prone techniques, such as TU-tagging, largely obsolete.

### 4.2. Determining the Subcellular Localization for Putative Kinocilial Proteins

GFP-tagged proteins have long been valuable tools for live imaging of protein localization [94,95]. This technique has proven particularly invaluable for the analysis of protein localization and function in zebrafish hair cells [8,31,96,97,98,99,100,101,102,103,104]. GFP-tagging has also been used extensively to assay protein localization to microtubules and cilia [68,105,106,107,108], including in hair cells [109]. Additionally, MAP6 family members have been studied in cultured cells and in vitro using fluorescent protein tags, including MAP6-GFP [87,110,111] and SAXO1- and SAXO2-GFP [80,81]. Importantly, the GFP tag does not seem to interfere with the normal function or localization of cilia-associated proteins. MAP6-GFP and endogenous MAP6 display similar enrichment in the proximal part of neuronal axons [112], and GFP-tagged versions of the Eps15 homology domain proteins EHD1 and EHD3 exhibit the same restricted localization to the proximal part of the primary cilium as the endogenous proteins [113]. Thus, GFP-tagging of cilial proteins is a powerful tool for analyzing protein localization in live cells.

### 4.3. Regionalization of the Hair Cell Kinocilium

Primary cilia are compartmentalized along their proximal-distal axis into functional domains marked by distinct protein compositions [114,115]. However, the regionalization of the hair cell kinocilium has received little attention. In this study, we expressed tagged versions of hair cell-enriched proteins Ankef1a, Odf3l2a, and Saxo2 in zebrafish to demonstrate their localization to the kinocilium of hair cells. Live imaging of these transgenic proteins also reveals their unique kinocilial localization patterns (Figure 1 and Figure 2, summarized in Figure 6). Like YFP-tagged tubulin, Ankef1a-GFP is uniformly distributed along the entire length of the kinocilium. Interestingly, Odf3l2a and Saxo2 exhibit somewhat complementary localization patterns, with Odf3l2a being restricted to the proximal-middle region and Saxo2 accumulating at the kinocilial tips. Together, our study is the first to demonstrate the cilial localization of these proteins in hair cells and lends support to the idea that the kinocilium is compartmentalized along its proximal-distal axis. It remains to be tested whether these domains are important for kinocilial structure and the sensory function of hair cells.

Based on our results, we propose that Ankef1a, Odf3l2a, and Saxo2 represent three examples of proteins with distinct kinocilial localization or retention signals. Analogous to how tubulin monomers line up in an orderly fashion, Ankef1a may contain a “queue-up” motif that prevents accumulation once all available spots are filled, ensuring a uniform distribution of Ankef1a along the axoneme. Odf3l2a may encode a “stop” signal that restricts localization to the middle-proximal compartment. Lastly, Saxo2 may contain a “keep walking” signal that allows the protein to accumulate at the plus ends of the axonemal microtubules. These putative localizations or sorting motifs may interact with subunits of the intraflagellar transport (IFT) machinery and/or with components of the transition zone or axoneme to produce these three distinct kinocilial localization patterns. Our results lay the foundation for future structure-function experiments to understand how these proteins are targeted to cilia and, together with loss-of-function models, to characterize their functional domains.

### 4.4. Saxo2 Promotes Formation of Curved and Truncated Kinocilia

Kinocilial length and morphology are normal in our *ankef1a* and *odf3l2a* transgenics. However, Saxo2-mKate2 overexpression promotes the formation of curved kinocilia. SAXO proteins are members of the microtubule-associated protein 6 (MAP6) family that stabilize microtubules in axons, cilia, and flagella [79,80]. Recent work shows that MAP6 and SAXO are microtubule inner proteins (MIPs) that can localize to the intraluminal space of microtubules [83,84,85,87]. Full-length MAP6 can bind to either the outside or inside of microtubules, and intraluminal localization requires an N-terminal cysteine-rich domain [87]. Luminal MAP6 can stabilize the formation of curved or helical microtubules, a property that externally bound MAP6 does not possess [87]. Saxo2 also features a cysteine-rich N-terminal domain of the MAP6 family [81]. Given that only luminal MAP6 induces microtubule curvature, it is possible that overexpressed Saxo2 also uses its N-terminal domain to localize to the luminal space of axonemal microtubules in hair cells. However, additional experiments are required to verify that the curvature observed in the Saxo2-mKate transgenics is Saxo2-dependent and to determine whether endogenous Saxo2 is a bona fide MIP in the kinocilium. However, our TU-tagging dataset is also significantly enriched for other known MIPs [116], including cilia and flagella-associated proteins (Cfap20, 45, and 52), EF-hand domain containing (Efhc) proteins 1 and 2, meiosis-specific nuclear structural 1 (Mns1), and tektins 1–3 (Appendix A). There are likely to be additional hair cell-enriched MIPs, and these data indicate a role for intraluminal microtubule-associated proteins in our senses of hearing and balance.

Saxo2 overexpression also promotes the formation of bulbed kinocilia, marked by the accumulation of Saxo2 at the tip. This distal accumulation is not accompanied by a concomitant redistribution of Ankef1a, Odf3l2a, or tubulin (Figure 5 and Appendix A). This is unlike the kinocilial bulb of bullfrog hair cells that labels strongly for beta-tubulin [117]. Thus, it is likely that the “canonical” and Saxo2-induced bulbs are formed by different mechanisms. We also found that Saxo2 overexpression promotes the formation of truncated kinocilia and that higher levels of Saxo2 positively correlate with more severe truncations (Figure 4). This correlation suggests that excess Saxo2 commandeers the IFT machinery, leading to Saxo2 accumulation at the tip, possibly at the expense of other components required for normal growth of the kinocilium. However, the kinocilial length is unchanged in the Saxo2 transgenics with more modest levels of expression (Saxo2-mKate2 “mild” and Saxo2-GFP). Our results differ from those of Dacheux et al., who reported that SAXO1 overexpression almost doubles the length of the primary cilium in cultured RPE1 cells [80]. This discrepancy could be due to differences in the SAXO proteins and/or cell types. We also noted differences between the Saxo2-GFP and Saxo2-mKate2 phenotypes. Although the tip-enriched distribution patterns are similar between the two forms, we did not observe curvature or truncation of kinocilia in the Saxo2-GFP line (Appendix A). However, since the Saxo2-mKate2 phenotype intensifies with higher expression levels, we suspect that the comparatively lower levels of Saxo2-GFP expression are the reason for the phenotypic discrepancy between the two tags. There is also reason to believe that the mKate2 version of Saxo2 is trafficked more efficiently to the kinocilium. Qualitatively, we observed a greater accumulation of Saxo2-GFP in the subcuticular zone relative to Saxo2-mKate2, and Saxo2-mKate2 tends to accumulate at the tips to the partial exclusion of Saxo2-GFP when they are co-expressed (Figure 5). Although the formation of bulbed and truncated kinocilia is likely an artifact of Saxo2 overexpression, the cellular phenotype is nonetheless interesting due to its lack of a behavioural correlate. The adult viability and qualitatively normal behaviour of the severe Saxo2 phenotype suggest that a partial disruption of kinocilial morphology is tolerated under laboratory conditions. This tolerance and possible genetic redundancy may account for why hair cell-enriched kinocilial genes have not been identified in genetic screens based on specific auditory or vestibular behavioural phenotypes [118].

### 4.5. Evaluation of ANKEF1, ODF3L2, and SAXO2 as Putative Hearing Loss Genes

Although we and others have identified many genes that may contribute to the formation and function of the hair cell kinocilium, we focused on *ankef1a*, *odf3l2a*, and *saxo2* in this study, based on preliminary reports of their genetic location in relation to mouse and human deafness loci. Mouse *Ankef1* is within the coloboma (*Cm*) deletion locus, the phenotypes of which include planar polarity defects in the cochlear epithelium, supernumerary hair cells, and hearing and balance defects [26,27]. Deletion of the Notch ligand gene *Jag1* is responsible for most of these hearing-related phenotypes, but phenotypical differences between *Cm* and *Jag1* heterozygotes leave open the possibility that other genes in the *Cm* locus contribute to the phenotype. Since ANKEF1 is a kinocilial protein, and Notch pathway mutants display polarity defects indicative of mispositioned kinocilia [119,120], it is possible that the combined heterozygous loss of *Jag1* and *Ankef1* reveals a subtle role for ANKEF1 in kinocilial development, which would not be apparent otherwise. Because ANKEF1 has protein–protein interaction ankyrin repeats and calcium-sensing EF-hand domains, one may speculate that ANKEF1 mediates protein–protein interactions in the kinocilium in response to changes in calcium concentrations.

A targeted knockout of *Odf3l2* in mice results in high-frequency hearing loss and vestibular deficits [70,121,122]. The cellular basis for these phenotypes is unclear, as mouse *Odf3l2* is not robustly expressed in either cochlear or vestibular hair cells [50,88,123]. However, *Odf3l2* expression is detected in outer pillar cells [123,124], suggesting a non-hair cell basis for the mutant phenotype. Interestingly, the homologous *Odf3b* gene is strongly expressed in mammalian hair cells, but knockout mice do not exhibit auditory or vestibular phenotypes [122]. Zebrafish *odf3l2a* is expressed in both inner ear and lateral line hair cells (Figure 1C) [57,125] whereas *odf3b* is specifically expressed in ciliated cells of the olfactory epithelia and pronephros [126]. Zebrafish also have a *odf3l2b* ortholog, whose expression is restricted to the testis [127]. Thus, vertebrate hair cells may utilize different *ODF3* orthologs: *ODF3B* in mammals and *odf3l2a* in fish.

*Saxo2* transcripts are enriched in mouse cochlear and vestibular hair cells [50,88,123,124], as well as in the inner ear and lateral line hair cells of zebrafish (Figure 1D) [57,125]. Enriched expression in the olfactory epithelia is also shared between mice and fish [107]. However, unlike *Odf3l2*, there are no published loss-of-function animal models of *Saxo2*. Human *SAXO2* (also known as *FAM154B*) and *ODF3L2* are reported in SHIELD as mapping to deafness loci *DFNA30* and *DFNB72*, respectively [28]. *DFNA30* is an autosomal dominant form of nonsyndromic hearing loss for which the causative gene remains unknown [77]. However, neither *SAXO2* nor *ODF3L2* resides within any known deafness loci, based on our analysis of the genetic markers used to map these and other nearby loci. Although it is still possible that *ODF3L2* and *SAXO2* contribute to auditory or vestibular function, we propose that they should be excluded as candidates for the currently known deafness loci.

### 4.6. Benefits and Limitations of This Study

In contrast to the short primary cilium of most cell types, sensory hair cells often possess prominent kinocilia, particularly the vestibular hair cells of the semicircular canals. This study takes advantage of the long kinocilium in the cristae of larval zebrafish and our ability to image tagged protein localization in live animals to understand more about the sub-cellular and sub-cilial localization of these proteins. The benefits of live imaging of fluorescently tagged proteins include avoiding both the morphological distortions caused by fixation and processing, and the inconsistencies of antibody staining. However, limitations include the possibility of altered protein localization patterns caused by the fluorescent protein tag, or from the use of a heterologous promoter to drive overexpression. However, the fact that all three proteins tested here display unique kinocilial and cytoplasmic distributions suggests that these tagged proteins reflect their endogenous localization patterns. An unexpected result of this study is Odf3l2a-GFP localization to the nucleus. Immunostaining of the endogenous protein is necessary to confirm this finding. If true, it presents the fascinating possibility of a communication channel between the kinocilia and the nucleus. Overexpression can also cause spurious phenotypes that may not be directly related to protein functions. For example, the bulbed and truncated kinocilia observed in the “severe” Saxo2-mKate2 transgenic line may be an overexpression artifact caused by the overloading of the IFT machinery with excess Saxo2. It is also possible that the kinocilial phenotypes in the *saxo2-mKate2* transgenics are artifacts of mKate expression since we did not observe similar phenotypes when Saxo2 is tagged with EGFP. However, the curved kinocilia we observed are consistent with the previously reported function of MAP6-family proteins, supporting the idea that transgenically expressed Saxo2 stabilizes microtubule curvature in this case.

## 5. Conclusions

In this study, we report the hair cell-enriched expression of zebrafish *ankef1a*, *odf3l2a*, and *saxo2* and show that tagged versions of these proteins exhibit distinct distribution patterns within the kinocilium. Together, our results suggest that the hair cell kinocilium is regionalized along its proximal-distal axis.

## Figures and Tables

**Figure 1 jdb-11-00028-f001:**
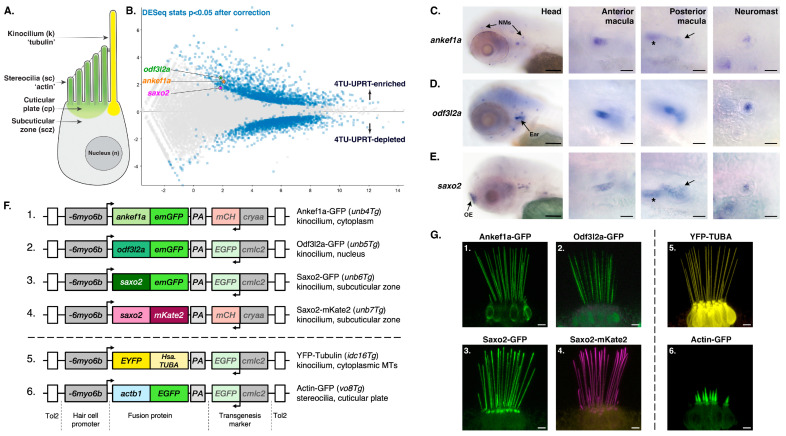
Identification of cilium-associated genes with enriched expression in zebrafish hair cells. (**A**) Diagram of a generic sensory hair cell. (**B**) Scatter plot of differentially expressed genes identified by TU-tagging. *ankef1a*, *odf3l2a*, and *saxo2* are indicated in the 4TU-UPRT-enriched population. (**C**–**E**) mRNA in situ hybridization (ISH) for *ankef1a*, *odf3l2a*, and *saxo2* in zebrafish larvae at 3 days post-fertilization (dpf). NMs = neuromasts; OE = olfactory epithelium. In the *ankef1a* and *saxo2* posterior macula panels, arrows point to the posterior macula sensory patch and asterisks indicate the out-of-focus anterior macula. (**F**) Schematic of transgenic constructs used in this study. (**G**) Overview of Ankef1a-GFP (1), Odf3l2a-GFP (2), Saxo2-GFP (3), Saxo2-mKate2 (4), YFP-TUBA (5), and Actb1-GFP (6) protein localization in the lateral cristae of larval zebrafish at 5–6 dpf. Scale bars are 100 µm for the headshots in (**C**–**E**), and 20 µm for all other ISH panels. Scale bars are 5 µm in (**G**).

**Figure 2 jdb-11-00028-f002:**
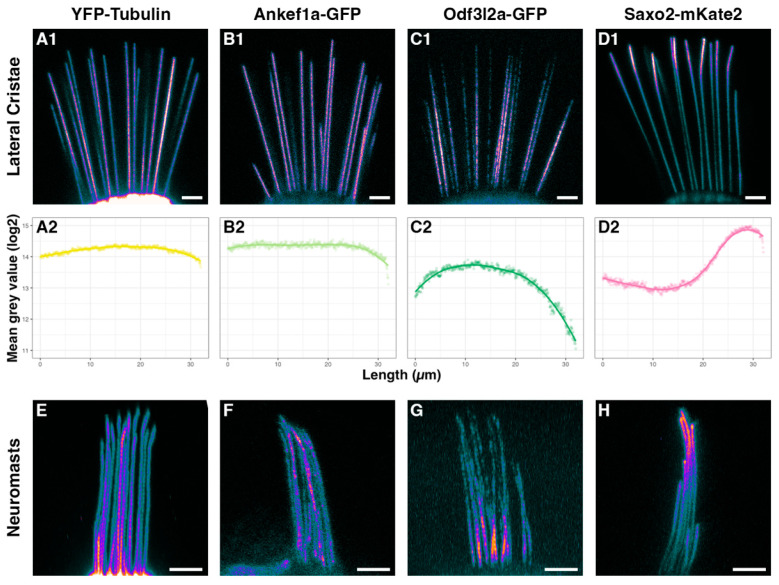
Distributions of tubulin, Ankef1a, Odf3l2a, and Saxo2 in the hair cell kinocilium at 5–6 dpf. (**A1**–**D1**) Pixel intensity-encoded images of tagged alpha-tubulin, Ankef1a, Odf3l2a, and Saxo2 proteins in the kinocilia of lateral cristae hair cells. Brighter colours indicate higher fluorescence intensity. (**A2**–**D2**) Scatter plots of averaged mean grey values (log2) for fluorescence intensity from kinocilia in the lateral cristae of YFP-TUBA (n = 35 kinocilia from 8 individuals), Ankef1a-GFP (n = 23 from 8 individuals), Odf3l2a-GFP (n = 30 from 11 individuals), and Saxo2-mKate2 (n = 17 from 6 individuals). Position 0 µm is from the proximal region near the base of kinocilia and position 32 µm is the distal-most tip. (**E**–**H**) Pixel intensity-encoded images of neuromast kinocilia from YFP-TUBA, Ankef1a-GFP, Odf3l2a-GFP, and Saxo2-mKate2 expressing zebrafish larvae. Scale bars are 5 µm in panels (**A1**–**D1**,**E**–**H**).

**Figure 3 jdb-11-00028-f003:**
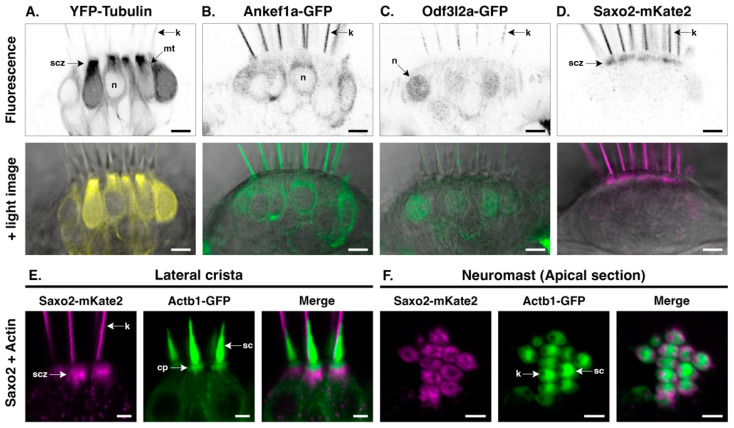
Distinct protein localization patterns in the hair cell soma. (**A**–**D**) Details of YFP-TUBA, Ankef1a-GFP, Odf3l2a-GFP, and Saxo2-mKate2 in the hair cell body at 5–6 dpf. The top panels are black-and-white representations of the fluorescence channel and the lower panels overlay the fluorescence channel with the corresponding light image. (**E**,**F**) Saxo2-mKate2 (magenta) accumulations in the apical part of the hair cell body in cristae and neuromasts. GFP-tagged beta-actin is shown in green. cp = cuticular plate; k = kinocilium; mt = microtubule; n = nucleus; sc = stereocilia; scz = subcuticular zone. Scale bars are 5 µm in (**A**–**D**) and 2.5 µm in (**E**,**F**).

**Figure 4 jdb-11-00028-f004:**
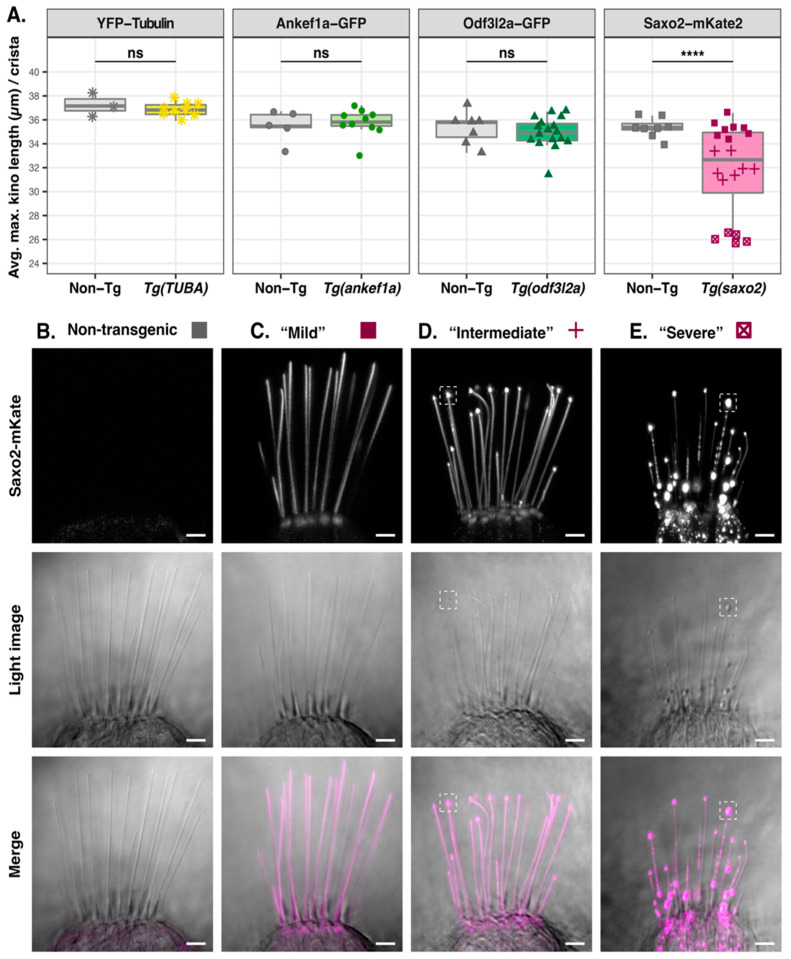
Effect of tubulin, Ankef1a, Odf3l2a, and Saxo2 overexpression on kinocilial length and morphology at 5–6 dpf. (**A**) Boxplots of maximum kinocilia length in the lateral crista of *YFP-TUBA*, *ankef1a-GFP*, *odf3l2a-GFP*, and *saxo2-mKate2* transgenics and their non-transgenic siblings. Data points are the average length of the five tallest kinocilia from an individual larva. Based on the morphological phenotypes of the three Saxo2-mKate2 transgenic lines, these data points are grouped as “mild” (solid squares), “intermediate” (crosses), and “severe” (boxed x’s). Two-tailed Welch’s *t*-test statistics—YFP-tubulin (6 dpf): non-Tg, n = 3 larvae, mean = 37.3 µm; *Tg(TUBA)*, n = 9, mean = 36.9 µm; t = 0.67757, df = 2.5116, *p* = 0.5551. Ankef1a-GFP (5 dpf): non-Tg, n = 5, mean = 35.5 µm; *Tg(ankef1a)*, n = 10, mean = 35.7 µm; t = 0.41384, df = 7.079, *p* = 0.6912. Odf3l2a-GFP (5 dpf): non-Tg, n = 7, mean = 35.3 µm; *Tg(odf3l2a)*, n = 17, mean = 35.0 µm; t = 0.5913, df = 10.931, *p* = 0.5663. Saxo2-mKate2 (5 dpf) one-way ANOVA (*F*_3,24_ = 193.3, *p* < 2 × 10^−16^). See text for results of pairwise comparisons between the Saxo2-mKate2 phenotypes. (**B**–**E**) Representative images of lateral cristae from non-transgenic (**B**) and *saxo2-mKate2* transgenics categorized as “mild” (**C**), “intermediate” (**D**), and “severe” (**E**). The dashed boxes in (**D**,**E**) highlight examples of the “bulbed” kinocilial phenotype. ns = not significant; **** = *p* < 0.00001. Scale bars are 5 µm.

**Figure 5 jdb-11-00028-f005:**
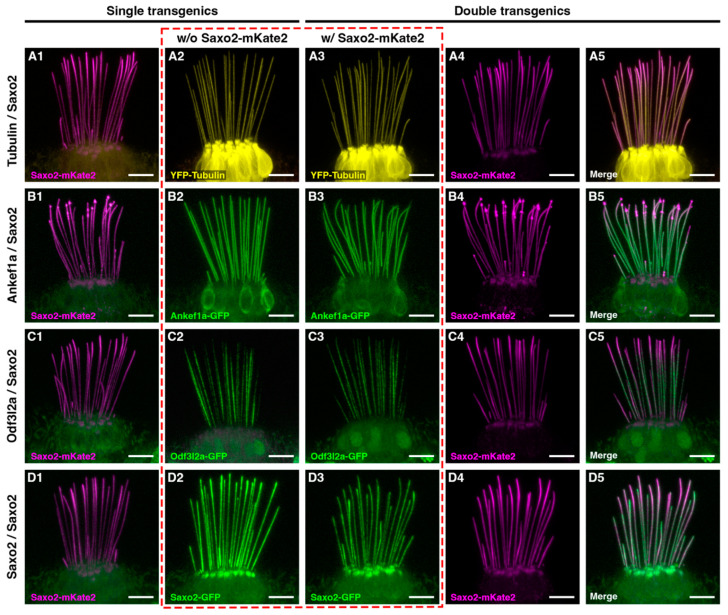
Co-expression of Saxo2-mKate2 with YFP-tubulin (**A**), Ankef1a-GFP (**B**), Odf3l2a-GFP (**C**), and Saxo2-GFP (**D**) at 5–6 dpf. Column 1 and 2 of each row shows single transgenic siblings, whereas columns 3 and 4 show individual channels from a double transgenic larva. Column 5 shows the merged channels from columns 3 and 4. The dashed red box compares YFP-TUBA, Ankef1a-GFP, Odf3l2a-GFP, and Saxo2-GFP localization without (**A2**–**D2**) or with (**A3**–**D3**) the presence of Saxo2-mKate2. Scale bars are 5 µm in all panels.

**Figure 6 jdb-11-00028-f006:**
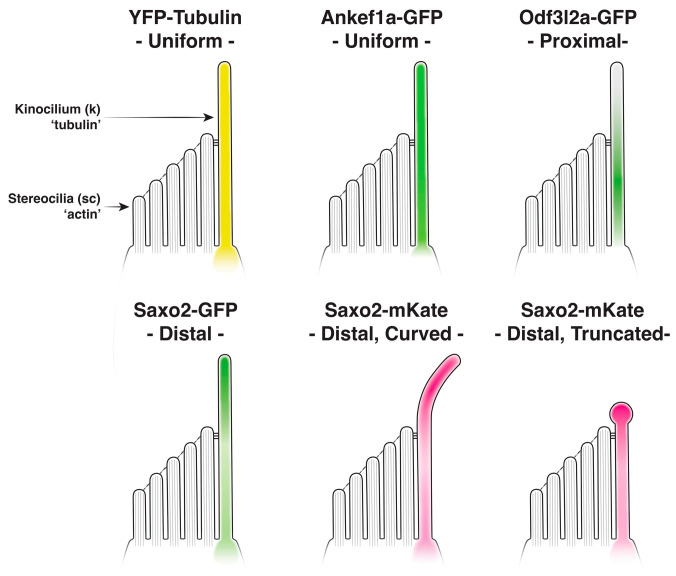
Summary of kinocilial localization patterns and phenotypes for Ankef1a, Odf3l2a, and Saxo2 transgenic proteins compared to tubulin.

## Data Availability

Data and code will be provided upon reasonable request from a qualified researcher.

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
