# Peer review of "Regionalized Protein Localization Domains in the Zebrafish Hair Cell Kinocilium"

_jdb, 2023, doi:10.3390/jdb11020028_

Round 1

Reviewer 1 Report

Paper summary:

In this study, the authors identify use TU-tagged RNA sequencing of zebrafish hair cells to identify genes important for hair cell kinocilia, an integral part of the mechanosensory hair bundle. After sequencing, the authors focus three genes ankef1a, odf3l2a, and saxo2 that have been implicated in hearing loss. They create new transgenic lines that fluorescently tag these genes of interest with emGFP or mKate2 to show that these proteins colocalize within the kinocilium. Using these transgenic lines, the authors demonstrate that Ankef1, Odf3l2a, and Saxo2 have distinct distribution patterns along the kinocilium as well as within the hair cell body. These distributions were compared to known tubulin- and actin-rich domains in hair cells. Finally, the authors also show that Saxo2-mKate2 expression alters kinocilium morphology, though it is unclear at this point if the phenotype is due to Saxo2 overexpression or is an artifact of mKate2 expression.

Overall, this study identified novel candidate genes to further explore kinocilium development, morphology, and function. The authors also generate new tools -transgenic lines- in which to study the function of these genes within living hair cells. The paper is well written, and the data and images are high quality. With some revision, I would recommend this manuscript for publication.

Major feedback:

1.     One thing that would be extremely helpful is images showing where GFP itself localizes in hair cells, by examining myo6b:GFP expression. Does this label also enter kinocilia? Even better would be to express a larger protein such as a tandem dimer of GFP. This type of localization analysis gives an important readout of where non-kinocilial proteins are permitted to diffuse. The main concern is with myo6:Ankef1a-GFP which has a diffuse localization in the cell and kinocilium.

2.     The authors make the claim that Saxo2 overexpression in the mKate2 fusion line results in the varying degrees of abnormal kinocilial morphology. This result is stated in the discussion and should be laid out more clearly in the results section. If the authors would like to relate expression levels with phenotype it is important to quantify the levels of Saxo2-mKate2 in the 3 different transgenic lines.

3.     It is unclear why Saxo2-mKate2 lines show a phenotype while the Saxo2-GFP line does not. Because levels of GFP and mKate2 cannot be directly compared, it is difficult to determine if the mKate2 tag or expression level is leading to the phenotype. If possible, the authors could transiently overexpress an untagged version of Saxo2 linked to a P2A GFP or transiently overexpress Saxo2-GFP and link expression levels to phenotype. If phenotypes are seen using either of these approaches it would link the phenotype to expression level of Saxo2 rather than an artifact of mKate2 expression. Alternatively, the authors can describe the concerns listened above and state that the experiment requires additional controls to truly verify that Saxo2 overexpression is responsible for the phenotype.

Minor feedback:

1.     Three of the first four words in the title seem redundant (i.e., regionalized, localization, and domains). Is there a more concise way to phrase this?

2.     The labels in Figure 1F are difficult to read. Please increase the text size.

3.     It would be helpful to have the transgenes labeled on the figure itself in Figure 1G.

4.     There are some hyphens and commas missing in the text.

5.     The “IFT” acronym is used before the acronym is defined in the text (see page 14). Please move the IFT definition to the first appearance of IFT in the text.

6.     The use of the present tense in parts of the results (when describing new data/results) is generally done in the past tense.

7.     Were the cDNAs amplified in this study deposited at NCBI?

8.     Line 91 is a myo6b missing from this transgene?

9.     Line 214 were these three genes chosen also in Table S4?

Author Response

Thank you for all your helpful feedback. Please see the attachment.

Reviewer 2 Report

In this manuscript the authors carry out a transcriptomic analysis of zebrafish hair cells to identify potential kinocilia genes that are important for hearing. They identify three such genes showing they do indeed have kinocilia localization and interestingly in the case of two of the genes appear to localize to distinct regions of the kinocilia. Additionally, overexpression of one of the three genes appears to alter kinocilia morphology. I believe this work provides important new insight about the potential importance of these three genes in hearing, however, I feel there are areas where important information about how the experiments were carried out is missing and other areas where claims are made that are not completely supported by the data as presented. My main concerns are detailed below.

It would be helpful if more information was provided about the imaging. The authors mention taking confocal z-stacks, it would be useful to know how many slices were in these stacks and the distance between slices. It would also be useful to have more information about which neuromasts were imaged.

In the results they mention manually searching the dataset for “putative kinocilia genes that may be associated with deafness phenotypes in mice and humans.” It would be helpful if more information was provided as to  what they were looking for to define a gene as being a putative kinocilia gene.

In Figure 2 the localization of the different markers does not always seem consistent in cristae versus neuromasts. This is particularly the case for Odf3l2a where it appears more basally localized in the neuromasts. Having the quantification for the neuromasts similar to what is shown for the cristae would make it easier to compare the two localization patterns.

In Section 3.4 of the results the author’s conclude that none of their genes have the same distribution pattern as YFP-Tubulin and none overlap with one another. While overall that certainly is true both Saxo2 and Tubulin appear to be concentrated in the subcuticular zone. I think that colocalization warrants mentioning. Also as they have double transgenic Tubulin/Saxo animals shown latter in the paper I think it would be useful to show images here as to how much the two do colocalize in that region.

In lines 519-520 of the discussion the authors note that “higher levels of Saxo2 expression are correlated with more severe truncation” however there does not appear to be any data looking at Saxo2 expression levels in the different transgenic lines to make that conclusion. In the images in Figure 4 it does look like there is more concentration of Saxo2 at tips in the more severe lines, but it is not clear that there is more expression overall. If the authors wish to state that conclusion they should have data to support it.

In lines 531-537 of the discussion the authors largely discount the lack of kinocilia morphology defects seen in the Saxo2:GFP line as being due to lower expression of Saxo2 in that line. As that is quite possibly true it’s also worth noting that the only line they see kinocilia morphology defects in is the only line that uses mKate, so I do not think some form of mKate artifact can be ruled out. This possibility should be brought up in the discussion.

Author Response

Thank you for all your valuable comments. Please see the attachment.

Reviewer 3 Report

This is an interesting study that identifies genes expressed in the zebrafish inner ear sensory maculae and neuromasts.  The authors characterise the expression patterns of transgenic fusion proteins corresponding to these genes in the hair cells of cristae in the inner ear.  Although some of the observations are very interesting, and the images are striking, there are a few inconsistencies in presentation of the data, where some additional information or clarity would help.  My main concern is with the comparison of the Saxo2-mKate2 fusion protein with the -GFP fusion proteins for the other two genes of interest.  It would be more helpful to present the -GFP forms for all three genes, and then go on to describe the aberrant kinocilial phenotypes induced by the Saxo2-mKate2 fusion proteins separately.  Detailed suggestions are listed below.

Line 2 – include the species (zebrafish) in the title.

Line 16 – the motivation for the study seems to be the relationship to human deafness loci – if this is the case it would be interesting to see some more figures showing the distribution of the proteins in the kinocilia of lateral line hair cells (as neuromasts are frequently used as models for hearing loss) in addition to those in the cristae, which have a vestibular function.

Line 19 – it would help to add information about the differences in expression patterns in the hair cell soma as well as the kinocilia to the Abstract.

Line 21 – Although there are few studies addressing this, regionalisation of the zebrafish hair cell kinocilium is not a new observation – see comment below. 

Line 222 – Comparison with the data presented in Ref. 44.  Which is the correct pattern?  Daniel and Panizzi (2019) also show hair cell expression, but show a broader expression overall.  Does the current study confirm the published whole embryo pattern?  It would be helpful to have pictures to show the whole embryo expression pattern for each gene (including the posterior lateral line neuromasts – are the genes expressed here as well?)  If the published and current patterns are different, what is the explanation for this?

Line 234 – if Ankef1a is enriched in ciliated cells, is it also expressed in other organs rich in cilia (brain, olfactory epithelium, pronephric ducts)? – again, an image showing expression in the whole embryo would help.

Line 239 – In situ expression at 3 dpf.  Are the genes expressed in the cristae? – these structures are not visible in the figure.  If not at this stage, do they become expressed in cristae at later stages?  As all the over-expression analysis is presented in the cristae, it is important to show whether or not the crista is a normal site of expression for the endogenous transcripts.

Line 242-244  What embryonic stages are shown in these panels? – this should be stated in the figure legend as well as in the text.

Line 325 – the Saxo2-GFP is confined to the supplementary data.  This should be presented in the main figure for ease of comparison with the other two GFP-tagged proteins.  As Saxo2-mKate2 gives an overexpression phenotype, this should be presented as a different figure, not as a direct comparison to the -GFP fusions for the other proteins.

Line 378 and Figure 4A, last panel.  It is not clear here if the three different phenotypes reflect variability within the same line, or whether they are each associated with a different independent line (as seems to be indicated in the text).  If this is the case, why group the data together in the same graph?  Are the data points in the other graphs (for the other fusion proteins) also taken from more than one independent line?

Line 402 – I cannot see the bidirectional mobility of the puncta in the video.  Details for how the video was acquired (speed of acquisition, number of frames, speed of playback) are needed in the Materials and Methods section, and/or in a legend to the video.

Line 407 – the summary here indicates that transgenic expression of Ankef1a-GFP and Odf312-GFP does not affect kinocilial height.  It is important to put the -GFP in the text here, especially as there is a difference between the Saxo2-GFP and the Saxo2-mKate2.  Saxo2-GFP should be added to the list, if it also does not affect kinocilial height.

Lines 409 and 524 – it is stated that lines with elevated levels of Saxo2-mKate2 have aberrant phenotypes, but this is not clear to me.  From the images shown, overall levels look comparable.  Expression levels should be quantified to show whether there is a correlation with severity of the phenotype; if no quantification is presented, this conclusion should be toned down.

Line 465 - Goodman and Zallocchi, J. Cell Sci. (2017) used immunohistochemistry to demonstrate the localisation of Pcdh15 and Itga8 to the tips of neuromast kinocilia in the zebrafish.

Line 478/Figure 6 – It would be helpful if this diagram could summarise the interesting data for the expression patterns in the soma as well, especially as this is pointed out as an interesting feature in the Discussion, i.e. the cytoplasmic expression for Ankef1a-GFP, the nuclear expression for Odf312a-GFP, and the sub-cuticular plate expression for Saxo2-GFP.  Saxo2-GFP should be shown, in addition to the two examples for Saxo2-mKate2.  (Fig. S1 appears to indicate that the kinocilia remain straight (no curved tips) when expressing Saxo2-GFP.)

Typos

Line 437 – previous (not previously)

Line 445 – inherent (not inherit)

Author Response

Thank you for your insightful suggestions. Please see the attachment.

Reviewer 4 Report

The authors performed a TU-trapping screen for genes enriched in the hair cells. They confirmed the enhanced expression of three genes in the hair cells by RNA in situ hybridization. They also examined the localization of fluorescently-tagged proteins both in the kinocilia and in the cytoplasm/nuclei. Finally, they found that overexpression of Saxo2-mKate2 leads to kinocilium defects. The manuscript is carefully written and the data are clearly presented. It is significant to identify and characterize new players in kinocilium formation and function. However, the manuscript in its current form is incomplete and readers cannot draw any concrete conclusions based on the data presented. The following improvement must be made to increase the rigor of the work.

major comments

1.     Since overexpression of Saxo2-mKate2 disrupts kinocilium formation, its localization cannot be presented as a proxy for normal Saxo localization. In fact, Saxo2-GFP localization presented in the supplemental figure is not exclusively distal. 

2.     A more fundament issue is whether the localization of the endogenous proteins can be extrapolated from observing the fluorescently labeled, overexpressed proteins. The authors need to examine the endogenous protein localization.

3.     It would be more convincing to provide another method of confirmation that these three genes are indeed enriched in hair cells, like qRT-PCR from purified hair cells compared to the whole embryo/larva. It is particularly necessary as the authors stated that a previous report was contradictory to what they reported here.

4.     The idea of compartmentation of the cilia is not new and there are proteins that mark different parts of the cilia. The authors should try to compare the localization of the three proteins to proteins that are known to mark different parts of the cilia.

5.     The authors found that two of the proteins are not within known hearing loss loci, and yet, they repeatedly claim that all three genes are “putative hearing loss” genes.  They should not emphasize this functional connection as they did not perform any functional test on these proteins.

minor issues:

1.     line 141: “washed with twice” should be “washed twice”

2.     line 437: “previously report” should be “previous report”

3.     line 445: “inherit” should be “inherent”?

4.     line 484 “an uniform” should be “a uniform”

Reviewer 5 Report

Going beyond previous work of the first author (ref 30), this study uses a modified TU incorporation method to identify mRNAs of additional novel proteins associated with the kinocilium of hairs cells in zebrafish larvae. Thereafter, the authors validate and further assess the localization of three new candidates where human orthologs map in the vicinity of genetic loci associated with hearing loss. In addition, live imaging of fluorescent protein fusion and their transgenic overexpression also reveal a novel gain-of-function phenotype of bulbed and curved kinocilia in a fraction of Saxo2-mKate2 larvae. Significant differences in the complementary localization of the three proteins examined suggest that also within kinocilia, distinct zones are enriched for distinct protein, and the three candidates examined will be fertile ground for future studies to further explore potential sensory functions in the inner ear.

The quality of the data is excellent, and the writing of the manuscript including the methods sections is very polished and easy to follow. Strengths and limitations of the study are also discussed clearly. Thus, I think the article merits publication even in its present form.

I have only minor questions about the interpretation of the Saxo2-mKate2 overexpression phenotype:

1) The authors suggest that the limited penetrance of this phenotype could mean that it may only manifest in those larvae that express the highest levels of the transgene. Assuming that there are probably no antibodies available to estimate protein levels, how does the mRNA expression level of the transgene compare to the levels of endogenous Saxo2 mRNA in hair cells?

2) Bulbing and curving of kinocilia was not observed in Saxo2-GFP transgenic larvae: While this striking difference is discussed clearly, I was wondering whether the penetrance of this phenotype increases in double transgenics that expressed both Saxo2-GFP and Saxo2-mKate2? If it didn't increase (or even decreased, as one might think when looking at the existing data), why didn't the authors discuss the possibility that the mKate2 fusion might result in a mild dominant negative effect that may be alleviated by overexpressing Saxo2-GFP?

3) Related to the previous point: Does the Saxo2-GFP fusion mimic the ring-like localization of Saxo2-mKate2 around the cuticular plate shown in Fig. 3F?

Author Response

Thank you for your comments, Please see the attachment.

Round 2

Reviewer 3 Report

The authors have dealt with my recommendations satisfactorily.

Reviewer 4 Report

The authors did not address any of my major concerns.